# Navigation of a Freely Walking Fruit Fly in Infinite Space Using a Transparent Omnidirectional Locomotion Compensator (TOLC)

**DOI:** 10.3390/s21051651

**Published:** 2021-02-27

**Authors:** Pikam Pun, Jacobs Brown, Tyler Cobb, Robert J. Wessells, Dal Hyung Kim

**Affiliations:** 1Department of Mechanical Engineering and Energy Processes, Southern Illinois University Carbondale, Carbondale, IL 62901, USA; pikam.pun@siu.edu; 2Department of Mechanical Engineering, Kennesaw State University, Marietta, GA 30060, USA; jbrow532@students.kennesaw.edu; 3Department of Physiology, Wayne State University, Detroit, MI 48201, USA; tyler.cobb@wayne.edu (T.C.); rwessell@med.wayne.edu (R.J.W.)

**Keywords:** motion compensator, tracking, *Drosophila melanogaster*

## Abstract

Animal behavior is an essential element in behavioral neuroscience study. However, most behavior studies in small animals such as fruit flies (*Drosophila melanogaster*) have been performed in a limited spatial chamber or by tethering the fly’s body on a fixture, which restricts its natural behavior. In this paper, we developed the Transparent Omnidirectional Locomotion Compensator (TOLC) for a freely walking fruit fly without tethering, which enables its navigation in infinite space. The TOLC maintains a position of a fruit fly by compensating its motion using the transparent sphere. The TOLC is capable of maintaining the position error < 1 mm for 90.3% of the time and the heading error < 5° for 80.2% of the time. The inverted imaging system with a transparent sphere secures the space for an additional experimental apparatus. Because the proposed TOLC allows us to observe a freely walking fly without physical tethering, there is no potential injury during the experiment. Thus, the TOLC will offer a unique opportunity to investigate longitudinal studies of a wide range of behavior in an unrestricted walking *Drosophila*.

## 1. Introduction

Animal behavior is complex and hard to understand because it is affected by various factors, such as the external environment. Recently, an automated system and robot have been integrated into biological experiments for a better understanding of animal behaviors [1,2,3,4,5,6]. Automated robotic tools allow researchers to observe animal behavior in controlled conditions by manipulating the animal or environment. A fruit fly, *Drosophila melanogaster*, is one of the most popular animal models to study behavior in the laboratory. The most common method to observe behavior in a walking fruit fly is using a custom-designed chamber with a limited spatial area [7,8,9,10,11]. To overcome the spatial limitation, simple mechanical devices have been implemented to alter the chamber’s position to induce behavior in a fruit fly, such as negative geotaxis [12,13,14]. Brain imaging in a freely walking fruit fly has been acquired within a specialized arena in the limited space [15]. However, because the animal behaves in the confined chamber, these types of experiments have been necessarily limited.

Other studies demonstrated behavioral observation with a tethered fruit fly [16,17,18,19]. A head-fixed fly walks on top of the air suspended sphere and its behavior is observed by measuring the rotation of the floating ball, which enables a fruit fly to walk in the virtual infinite space. The optical brain imaging with the tethered fly which walks on a ball allows observing behavior and brain activity simultaneously [20,21]. However, physical restriction by tethering could hinder the natural behavior of a fruit fly due to the potential risk of injury [22,23,24,25] even though the tethering may be inevitable for brain imaging. Virtual reality has been integrated with these systems to create a realistic environment [19,26]. Experiments with tethered insects have been carried out for other insects such as ants [27] and crickets [28]. To eliminate the risk of injury, locomotion compensators with a sphere have been utilized for various insects such as a fruit fly [29], honeybee [30], a pill bug [31], and a moth [32]. Even though the locomotion compensators track behavior in insects without tethering, it is still challenging to integrate additional apparatus such as a microscope, due to the spatial limitation around the device.

Even though the behavior of a fruit fly has been successfully investigated for a long time, a behavioral study of a completely mobile fruit fly remains challenging due to the spatial limitation and the complexity of the external environment. Because animal behavior is affected by various factors in the external environment, it is hard to observe animal behavior realistically.

In this paper, we developed the Transparent Omnidirectional Locomotion Compensator (TOLC), which compensates movements of a freely walking fruit fly and measures its behavior without tethering and spatial limitation. The TOLC compensates not only a position but also the heading of a freely walking fruit fly. The inverted imaging system of the TOLC secures a space for additional experimental apparatus. We also demonstrated the practical functionality of the TOLC via phototaxis behavioral experiments in *Drosophila*. Because observation without tethering prevents a fruit fly from damages or injuries, the TOLC will offer a unique opportunity to investigate longitudinal studies of a wide range of behavior in an unrestricted walking *Drosophila*.

## 2. Methods

### 2.1. Overview of Transparent Omnidirectional Locomotion Compensator (TOLC)

The overall schematic drawing of the TOLC is illustrated in Figure 1a. The design of the TOLC is similar to that of a balancing robot, a mobile robot that navigates with a single spherical ball only [33,34,35]. The design of the TOLC is an inverted version of the balancing robot. While a fruit fly is walking on top of the ball, the TOLC compensates a fly’s motion by rotating a transparent sphere (Precision Plastic ball Company, Franklin Park, IL, USA, 14179) using three omnidirectional wheels (Nexus Robot, Hongkong, China, 100 mm diameter). The size of the sphere is 4 inches in diameter and is located on top of three omnidirectional wheels illustrated in Figure 1a. The omnidirectional wheel is designed to achieve continuous contact with a sphere using alternated passive rollers [36], which creates a nearly continuous rotation of the sphere while translational motions are minimized. Three omnidirectional wheels were placed equally at every 120° and oriented at 40° with respect to the *xy*-plane to support the sphere. The omnidirectional wheels are directly attached to the DC servomotors (Robotis, Lake Forest, CA, USA, MX-64AT) to minimize a mechanical backlash. The near-infrared (NIR) camera underneath the transparent sphere records a motion of a walking fruit fly while light-emitting diodes (LEDs) illuminate a fly motion from the bottom of the sphere. Two optical encoders (Logitech, Newark, CA, USA, G502 HERO) were placed at the mid-plane of the sphere to measure the rotations of the sphere in the *x*-, *y*- and *z*-axis. To prevent a fruit fly from flying, we created a fly chamber on top of the sphere with a limited height (2 mm) using the cover glass (Brain Research Laboratory, Waban, MA, USA, 4550-1D) to let a fly walk in the chamber. Because the fruit fly is kept at the top of the sphere most of the time while tracking, the height of the chamber can be maintained during the experiment. Figure 1b,c shows the overall assembly drawing of the TOLC. Figure 1d,e shows a picture of the real system and the walking fly in the chamber of the TOLC, respectively.

### 2.2. Behavior Imaging System

Because a fruit fly is walking on top of the transparent sphere, the behavior camera underneath the sphere is able to capture a fly motion, which is similar to the inverted microscope. The advantage of this inverted imaging system is the secured space around and above the behavior chamber where an additional apparatus can be integrated for various future experiments with an external stimulus such as olfactory, vision, thermal stimuli. In addition, we utilized the 850 nm NIR LEDs (Osram, Wilmington, MA, USA, SFH 4655-Z) to minimize visual distraction for a walking fruit fly. The arrays of the NIR LEDs are located around the sphere as illustrated in Figure 2. The NIR LEDs illuminate a walking fruit fly with an oblique angle from the bottom. Figure 2b shows a side view of the simulation of ray tracing. Figure 2c shows detailed simulation results at the behavior chamber where a fruit fly walks. Because the rays diverge laterally while the refracted light by a walking fly is captured by the camera sensor, this creates a high contrast dark-field image. The NIR camera (FLIR, Wilsonville, OR, USA, CM3-U3-31S4M-CS) with the lens (Tamron, NY, USA, 12VM1040ASIR) and the bandpass filter (Thorlab, Newton, NJ, USA, FB850-40) are located on the bottom of the transparent sphere to image behavior of a single fly. To minimize the motion blur, the exposure time was set to less than 1.0 ms with the synchronized LED illumination and the images were captured at 200 Hz.

The sample NIR image is shown in Figure 3a, whose field of view is 16.4 mm by 9.66 mm (640 by 376 pixels) with a pixel distance of 25.7 μm/pixel. To detect the position and orientation of a walking fruit fly, the captured image is processed in real-time with the following step. First, we applied the Gaussian filter on the image and binarized the filtered image with the threshold that keeps the same fly size. Then, we find the centroid and orientation of the fly object from the binary image. The configuration of a fly in the camera coordinate is defined by *p_cam_* = [*x_cam_*, *y_cam_*, *ϕ_cam_*], where *ϕ* is the orientation of the fly’s heading with respect to the *x*-axis in the image shown in Figure 3b.

The imaging system includes the transparent ball as a spherical ball lens, which may cause image distortion. To measure the image distortion, we captured the image of the concave pattern, which makes full contact with the surface of the sphere using the stereolithography (SLA) 3D printer (Creality, Shenzhen, China LD-002R). Figure 4a shows the captured images of the patterns, in which the distance between patterns is 1 mm in both the *x*- and *y*-axis. The yellow dots are the centroids of the patterns in the captured image. We evaluated the distortion with those centroids [37], and Figure 4b shows the undistorted image of the pattern and the red dots are the centroids of the pattern. The dotted circle in Figure 4a,b shows this 4 mm radius circle from the image center. At the center of the image, the differences between the centroids of the patterns in the original (yellow dots) and the undistorted image (red dots) are negligible. However, those at the corner of the image are noticeable. The result distortion in Figure 4b is the barrel (or negative) distortion, which is typical in a zoom lens. To quantify the distortion, we defined the distortion as follows:(1)$$\mathrm{Distortion}\;\left(\%\right)=\left(H^{*}-H\right)/H\times 100$$
where the *H* and *H** are the distances of the point on the image from the center in the undistorted image and the original image, respectively. Figure 4c shows the distortion rate of the imaging system. The distortion at the radial distance from the center of 4 mm is −0.32%, which equals 12.8 μm or 0.5 pixels in the image. At the corner of the image (the radial distance from the center of 9.58 mm), the distortion is 1.89%, which equals 7.0 pixel or 181 μm. We assumed that there is no image distortion because the image distortion is not significant (<1 pixel) inside of the 4 mm circle where a freely walking fly is mostly staying (less than 2 mm error range for 99.3% of experiment time, see the discussion section) during tracking.

### 2.3. System Modeling

Figure 5a,b show the kinematics of the TOLC [32,38]. We assumed that the transparent sphere rotates only without translational motion. Let *θ_sp_* denotes the angles of sphere rotations in *x*-, *y*-, and *z*-axis, and *θ_w_* represents the angles of three omnidirectional wheel rotations. The angular velocity *ω_sp_* (= ${\dot{\theta }}_{sp}$) is related to the angular velocity of three omnidirectional wheels *ω_w_* (= ${\dot{\theta }}_{w}$) as
(2)$$\omega_{sp}=M\omega_{w}$$
where M is the matrix, which is identified based on the configuration of the TOLC in Figure 5. The matrix M is expressed as
(3)$$M=\frac{r_{w}}{r_{sp}}\left[\begin{array}{ccc} \sin\alpha \cos\beta_{1} & \sin\alpha \cos\beta_{2} & \sin\alpha \cos\beta_{3}\\ -\sin\alpha \sin\beta_{1} & -\sin\alpha \sin\beta_{2} & -\sin\alpha \sin\beta_{3}\\ \cos\alpha & \cos\alpha & \cos\alpha \end{array}\right]$$
where *r_w_* is a radius of a wheel, *r_sp_* is a radius of a sphere, *α* is the oriented angle of omnidirectional wheels with respect to the *xy*-plane, and *β* is the angle between omnidirectional wheels. We set *α* = 40°, and *β* = [0, π/3, 2π/3]. The radius of the omnidirectional wheels *r_w_* is 50 mm, and the radius of the transparent sphere *r_sp_* is 50.8 mm.

The equation of motions of the TOLC, described in Figure 5, is as follows:(4)$$I{\ddot{\theta }}_{sp}+D{\dot{\theta }}_{sp}=U_{sp}$$
where *I* is the moment of inertia of the sphere, *D* is the damping coefficient, and *U_sp_* = [*u_x_*, *u_y_*, *u_ϕ_*] is the input. The control input *U_sp_* is converted to control input of three omnidirectional wheels *U* using the matrix M shown in Equation (3) as follows:(5)$$U_{sp}=MU$$

For the tracking control, the position error of a fly *e_fly_* = [*e_x_*, *e_y_*, *e**_ϕ_*] is defined as a distance between a fly position *p_cam_* from the center of the image *p_c_* = [*x_c_*, *x_c_*, *ϕ_c_*] as follows.
(6)$$e_{fly}=p_{cam}-p_{c}$$
which is also aligned with the top plane of the sphere shown in Figure 5a. The error *e_fly_* in image coordinate then converted to the error *e_sp_* = [Δ*θ_x_*, Δ*θ_y_*, Δ*θ_z_*] = [*e_y_*/*r_sp_*, −*e_x_*/*r_sp_*, *e**_ϕ_*] in the sphere coordinate, which is the amount of the rotation of a sphere required to compensate for the current position error. Then we computed the control input *U_sp_* = [*u_x_*, *u_y_*, *u_z_*] from on the error *e_sp_* using the proportional-integral-derivative (PID) control. The overall process is repeated for every image, and the average computational time, including image processing and computing control inputs, takes less than 2 ms, which is sufficient to run the tracking system at 200 Hz.

### 2.4. Tracking Path Generation

A travel path of a walking fly is reconstructed based on a fly position *p_cam_* in the camera coordinate and a position of the sphere *p_sp_* in the virtual *xy*-plane. A virtual sphere position *p_sp_* is defined as [*x_sp_*, *y_sp_*, *ϕ_sp_*], which shows a path of the top point of the sphere, where a fly is walking. The optical encoders measure the rotation changes of the sphere Δ*θ_sp_* = [Δ*θ_x_*, Δ*θ_y_*, Δ*θ_z_*] in the *x*-, *y*- and *z*-axis and then converted them into the position and orientation changes in the virtual plane Δ*p_sp_* = [Δ*x_sp_*, Δ*y_sp_*, Δ*ϕ_sp_*] = [*r_sp_*Δ*θ_y_*, −*r_sp_*Δ*θ_x_*, Δ*θ_z_*]. The virtual path of the sphere *p_sp_* is updated by adding the change of the virtual position into the current virtual sphere position as follows.
(7)$$p_{sp,i+1}=p_{sp,i}+R\Delta p_{sp}=p_{sp,i}+\left[\begin{array}{ccc} \cos\left(\phi_{sp,i}\right) & -\sin\left(\phi_{sp,i}\right) & 0\\ \sin\left(\phi_{sp,i}\right) & \cos\left(\phi_{sp,i}\right) & 0\\ 0 & 0 & 1 \end{array}\right]\Delta p_{sp}$$
where *p_sp,i_* and *p_sp,i+_*_1_ are the virtual sphere positions in the current and next step respectively. The matrix *R* is the rotational matrix with the current heading of the sphere *ϕ_sp,i_* in the virtual coordinate. The virtual fly position *p_fly_* = [*x_fly_*, *y_fly_*, *ϕ_fly_*] is computed by adding the virtual sphere position *p_sp_* and a fly position *p_cam_* using Equation (8). The virtual path of a walking fly is a sequence of *p_fly_*, which shows a travel trajectory of a walking fly in the infinite space.
(8)$$p_{fly}=p_{sp}+\left[\begin{array}{ccc} \cos\left(\phi_{sp}\right) & -\sin\left(\phi_{sp}\right) & 0\\ \sin\left(\phi_{sp}\right) & \cos\left(\phi_{sp}\right) & 0\\ 0 & 0 & 1 \end{array}\right]p_{cam}$$

## 3. Result and Discussion

### 3.1. Drosophila Melanogaster

We obtained wild-type Canton S from the *Drosophila* stock center (Bloomington, IN, USA) and used them for our experiments in this paper. All flies were raised in standard food (Genesee Scientific, San Diego, CA, USA, 66-117) bottles or vials at 25 °C in a 12-h dark–light cycle. The flies were collected from the vial when they are between 20 to 40 days old and transferred to the device under CO_2_ anesthesia. A fruit fly normally begins to move within a few seconds after transfer and the tracking device begins to track a fruit fly and compensate its motion immediately. Flies were left for five minutes on the device in order to habituate them before collecting the experimental data. All the experiments were performed in a dark enclosed box. The light source for the phototaxis experiment was white LEDs with a light intensity of 0.09 mW/cm^2^ measured using the optical power meter (Thorlabs, Newton, NJ, USA, PM100D).

### 3.2. Step Response to the Control Input

Figure 6 shows the response of the angular velocity of the sphere in the *x*-, *y*- and *z*-axis. These step responses were measured when the input is (a) *U_sp_* = [1, 0, 0], (b) *U_sp_* = [0, 1, 0] and (c) *U_sp_* = [0, 0, 1], respectively. The output angular velocity *ω_sp_* of the sphere was measured using optical encoders. To remove the noise, we applied a low-pass Gaussian filter to the output and averaged multiple measurements. In Figure 6, the final angular velocity of all three responses is reached at around 100 ms. The final angular velocity in the *x*- and *y*-axis in Figure 6a,b are faster than that in the *z*-axis in Figure 6c even though the magnitudes of the step inputs are the same. In Figure 6a, there are still some transient responses of *ω_y_* and *ω_z_* even though those become zero in steady-state response. The same transient responses of angular velocities in other axes are found as well in Figure 6b,c. Because two or three motors should be actuated simultaneously to apply the step input *U_sp_* as described in Equation (5), the different transient responses of each motor can cause transient responses in other axes.

### 3.3. Tracking of Unstimulated Behavior

To demonstrate the performance of the developed TOLC, a walking fruit fly was tracked on the device without any external stimulus in the dark chamber. The locomotion of the freely walking fruit fly was measured for five minutes. The regenerated trajectory in the virtual coordinate shows the overall path, whose total travel distance of a fly was 957 mm (Figure 7a). The position and orientation during the experiment in Figure 7b show that the fly walks and stops spontaneously on the TOLC. Figure 7c shows the trajectory information of the single bout illustrated in the red inset box in Figure 7a. During this bout, the fly moved forward and turned right (clockwise), and the tracking system compensated its error to maintain the fruit fly position in the field of view. The NIR images of the tracked fly at each time point is shown in Figure 7d. Once the fly walked, the TOLC system began to compensate its position and orientation error and followed behind the fly (Figure 7e–g) because our feedback control system relies on the error only. There are some fluctuations of the heading of the fruit fly in Figure 7g because the body of a fruit fly oscillates when it walks.

Figure 8 shows the error *e_fly_* during a single bout movement in Figure 7d. The distance error in Figure 8a shows that the error increases once the fly moves, and the error is kept around 0.7 mm after the TOLC begin to cancel the error. When the fly accelerated at around 0.7 s, the error in the *x*-axis increased, but it decreased back after the system compensated the error. In Figure 8b, the error *e_x_* is normally larger than *e_y_* because a fly mostly walks forward and the TOLC controls a heading angle toward the positive direction of the *x*-axis in the image coordinate. Thus, an error normally increases more along its longitudinal direction (*e_x_*) than its lateral direction (*e_y_*) during motion. In Figure 8c, the heading error increased when a fly turned at around 1.1 s, but the error decreased after the system compensated the error. At the end of the movement, the position and heading error decreased to zero.

### 3.4. Positive Phototaxis of Drosophila

Normally, a fruit fly moves toward the light, called positive phototaxis [39]. We observed live behavior during positive phototaxis in a freely walking fruit fly to demonstrate the potential use of the TOLC. In a dark chamber, LEDs emit broadband white light either on the left or right side of the fly walking direction as illustrated in Figure 9a. The location of LEDs is 183 mm apart from the fly behavior chamber and maintained because the TOLC maintains a heading of the walking fruit fly toward the same direction. Before the experiment, the fruit fly is placed on the TOLC in the dark chamber for five minutes. Then, the LEDs were turned on and toggled between the left and right sides every 30 s. Figure 9b shows the heading angles *ϕ_fly_* for 180 s. The fly turned and moved to the right (clockwise) for the first 30 s because the LEDs on the right side were on while those on the left side were off. After 30 s, the left LEDs turned on while the right LEDs were off. The fruit fly switched its direction to the left (counterclockwise) where when LEDs light toggled. During the experiment in Figure 9, the total travel distance is 1430 mm with an average speed of 12.38 mm/s, which is measured when it is walking only. This experimental data clearly shows the positive phototactic movement of an adult fruit fly.

## 4. Discussion

The range of the average velocity of a walking fruit fly on the TOLC is from 10 to 40 mm/s, where it is a similar range of the walking velocity in the previous studies [40,41]. The overall average walking velocity is 14.2 ± 4.20 mm/s, which is measured from 270 min of data across 20 flies in total. We measured the mobility, the ratio between bout (walking) and rest (interbout), and it shows that a fruit fly moves actively 50.4 ± 17.13% of the time on the TOLC. We assumed that a fly is at rest when the position change is less than one pixel which is equal to 5.08 mm/s to eliminate the noise. Figure 10a,b show the relationship between the distance error (*e_x_*^2^ + *e_y_*^2^)^1/2^ and the speed of a walking fruit fly on the TOLC. The distance error increases when a fly moves faster (Figure 10b). However, the distance error is mostly less than 1.0 mm regardless of the velocity of a walking fly. The distribution in Figure 10a shows that a fly is normally walking less than 20 mm/s, and the TOLC maintains its position within a 1.0 mm distance error.

Figure 10c shows the cumulative probability of distance error of walking fruit flies. In Figure 10c,d, the error data during entire experiments (blue) is higher than that while a fruit fly is in motion (orange) because error becomes almost zero when a fruit fly stops between walking bouts. We found that the distance error is less than 1 mm for 90.3% of the experiment time and less than 1.25 mm for 95.9% of the experiment time. While the fruit fly is in motion, the position error was less than 1 mm for 82.4% of the experiment time and less than 1.25 mm for 92.4% of the experiment time. Even though we exclude the data when a fruit fly is at rest, the system still keeps a position error less than 2 mm for more than 99.3% of the time. Similarly, Figure 10d shows the cumulative probability of heading error *e_ϕ_* of walking fruit fly. The heading error was maintained less than 5° for 80.2% of the experiment time and less than 8° for 90.7% of the experiment time. While a fruit fly is in motion, the heading error was maintained less than 5° for 68.8% of the experiment time and less than 8° 85.0% of the experiment time.

## 5. Conclusions

Here, we demonstrate the utility of the transparent omnidirectional locomotion compensator (TOLC) for tracking behavior in a freely walking fruit fly. The inverted imaging system with the transparent sphere secures the space around and above the chamber, where an additional apparatus such as virtual reality, argument reality, and the optical microscope can be integrated into the TOLC system. In the TOLC system, a fruit fly walks on the transparent sphere while the sphere rotates in the opposite direction of the movement to cancel out its motion. Because three omnidirectional wheels and servomotors control rotations of the sphere, the TOLC system can compensate not only the position but also the heading of a fly simultaneously.

The results have demonstrated that a fly can walk freely at 14.2 mm/s speed on average on the TOLC system, and the ratio of active walking is greater than 50% of the time. The TOLC system maintains the position and heading error of a freely walking fly less than 1 mm for 90.3% of the time and 5° for 80.2% of the time, respectively. An advanced control method such as Model Predictive Control [42] can improve the tracking performance of the TOLC system in the future. In addition, the phototaxis behavior of a freely walking fruit fly has been demonstrated to validate the potential utilization of the TOLC system in various behavioral experiments.

The TOLC system allows us to observe a freely walking fly without physical tethering. Thus, there is no potential injury during the experiment due to tethering, which is ideal for long-term research for multiple days and weeks. Additionally, we remain future work to improve the tracking performance of the TOLC system and integrate it with other apparatus such as an optical microscope for internal organ and brain imaging while behaving.

## Figures and Tables

**Figure 1 sensors-21-01651-f001:**
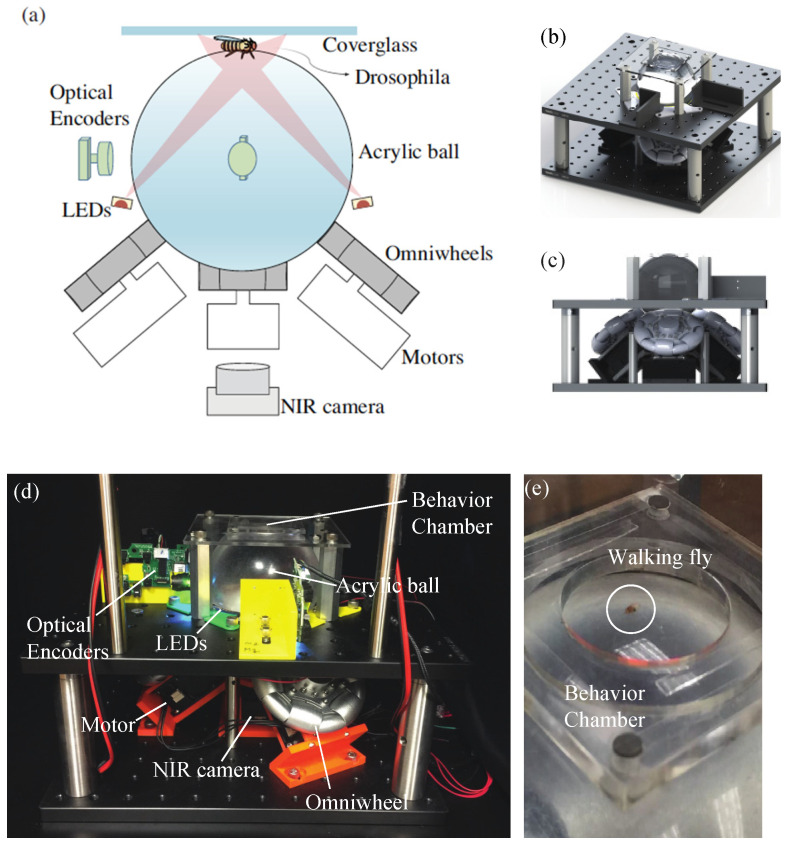
(**a**) Schematic drawing of the Transparent Omnidirectional Locomotion Compensator (TOLC) for a freely walking fruit fly (**b**,**c**) Design of Experimental set-up (**b**) isotropic view (**c**) front view (**d**) A real picture of the developed experiment device, the overall size is approximately 304.8 × 304.8 × 207 mm (width × depth × height) (**e**) A walking fruit fly on the real experimental set-up (see Appendix A).

**Figure 2 sensors-21-01651-f002:**
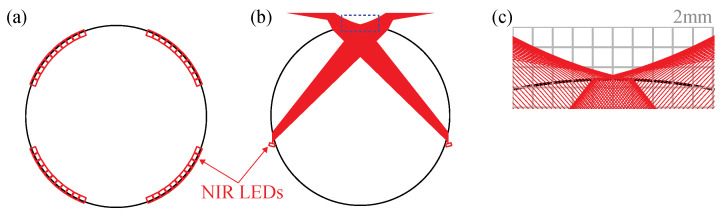
The illumination system for behavior imaging. (**a**) The top view of the NIR LEDs (red rectangle), 40 LEDs (a set of 10 LEDs in every 90°) are installed around the sphere. (**b**) The side view of the illumination system and simulation of ray-tracing (red lines) of 2 LEDs only shown. (**c**) The detailed results of ray-tracing simulation in the dotted box in (**b**).

**Figure 3 sensors-21-01651-f003:**
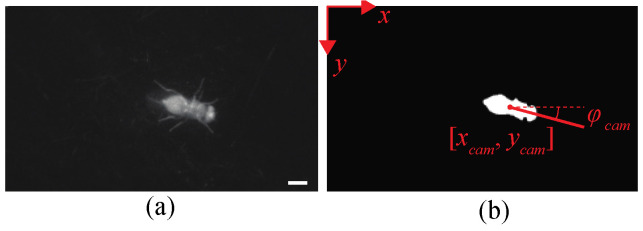
(**a**) An example of a captured NIR image of a walking fruit fly using the illumination system. The scale bar is 1 mm. (**b**) binarized image of the image in (**a**). The configuration of a walking fruit fly on the image *p_cam_* = [*x_cam_*, *y_cam_*, *ϕ_cam_*], which are the position in the *x*- and *y*-axis (red dot), and the orientation with respect to the *x*-axis (red line). The origin of the image coordinate is located on the left top corner. The pixel distance of a NIR image is 25.7 μm/pixel.

**Figure 4 sensors-21-01651-f004:**
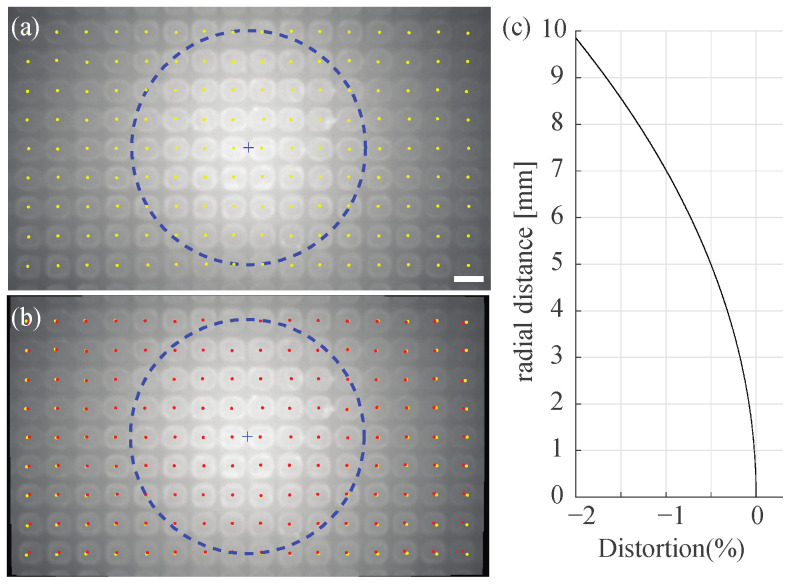
Image Distortion. (**a**) The original image of the 1 mm spaced pattern. Yellow dots are the centers of each object. The scale bar is 1 mm. (**b**) Undistorted images from the image in (**a**). The red and yellow dots are the centers of each object in the undistorted images and the original images, respectively. The radius of the blue dotted circle in (**a**,**b**) is 4 mm. (**c**) The measured distortion rate of the imaging system.

**Figure 5 sensors-21-01651-f005:**
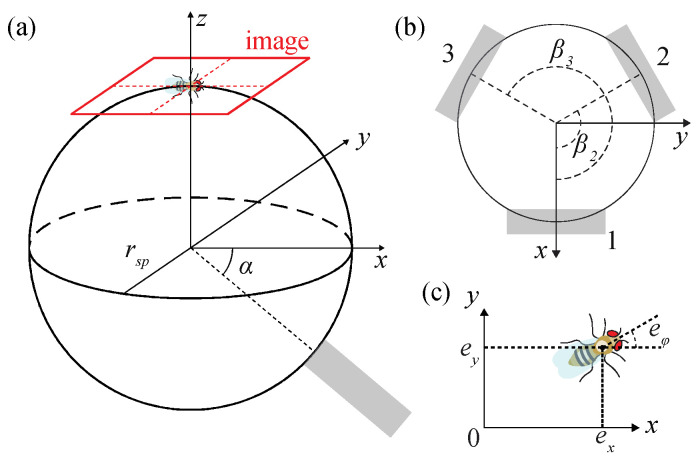
Kinematics of the TOLC system. (**a**) The overall system. The sphere is a transparent acrylic sphere and a gray block is an omnidirectional wheel. Only one wheel among three wheels is shown. The red rectangle is the field of view of a behavior image camera. *α* is of the omnidirectional wheels respected to the *xy*-plane. *r_sp_* is a radius of a transparent sphere. (**b**) The top view of the system. *β* is the angle of the omnidirectional wheels 1, 2, and 3 with respect to the *x*-axis. (**c**) Error of the walking fruit fly on the image *e_fly_* (*e_x_*, *e_y_*, *e**_ϕ_*), which is the distance from the center of an image. The origin is the center of an image (not shown in here). The coordinate of a NIR image aligns the sphere coordinate.

**Figure 6 sensors-21-01651-f006:**
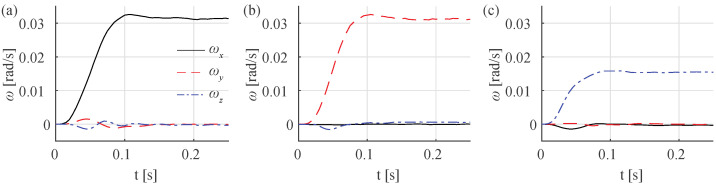
Step input response of the system. The step inputs are (**a**) *U_sp_* = [1, 0, 0], (**b**) *U_sp_* = [0, 1, 0] and (**c**) *U_sp_* = [0, 0, 1], respectively. The black solid line, red dashed line and blue dash-dotted line are the angular velocity *ω* in *x*-axis, *y*-axis and *z*-axis respectively.

**Figure 7 sensors-21-01651-f007:**
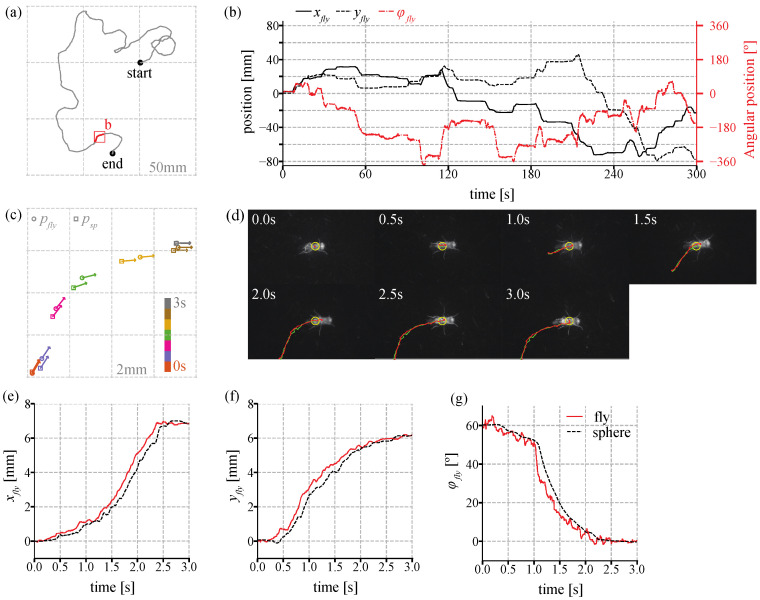
The behavior of the freely walking fruit fly on the TOLC system (**a**) An overall trajectory of a fly for 5 min. The grid size is 50 mm. The total travel distance is 957 mm. (**b**) The coordinates (*x**_fly_*, *y**_fly_*, *ϕ**_fly_*) of the walking fly during in the virtual coordinate. (**c**) The trajectory of *p_fly_* and *p_sp_* for 3 s which is shown as the red trajectory in (**a**). A circle and square makers show the position of the fly and the sphere in virtual coordinate, respectively. An arrow shows its heading. The data are displayed every 0.5 s. The grid size is 2 mm. (**d**) The NIR behavior images at each time point in (**c**). See Appendix A. The red and green are the trajectory of the walking fly and the sphere. (**e**–**g**) positions *p_fly_* and *p_sp_* (**e**) *x*-axis, (**f**) *y*-axis, and (**g**) heading *ϕ* of the walking fly. The solid red and dotted black lines are positions of the walking fly and the TOLC, respectively.

**Figure 8 sensors-21-01651-f008:**
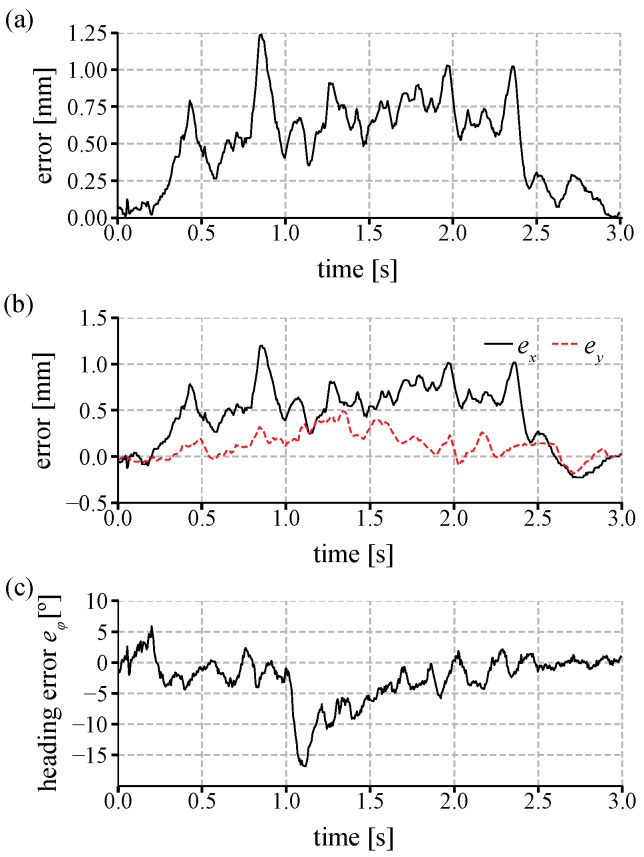
The error during the single bout in Figure 7c. (**a**) The overall distance error $e=\sqrt{e_{x}^{2}+e_{y}^{2}}$, (**b**) The errors *e_x_* and *e_y_* (**c**) the heading errors *e_ϕ_* in the image coordinate.

**Figure 9 sensors-21-01651-f009:**
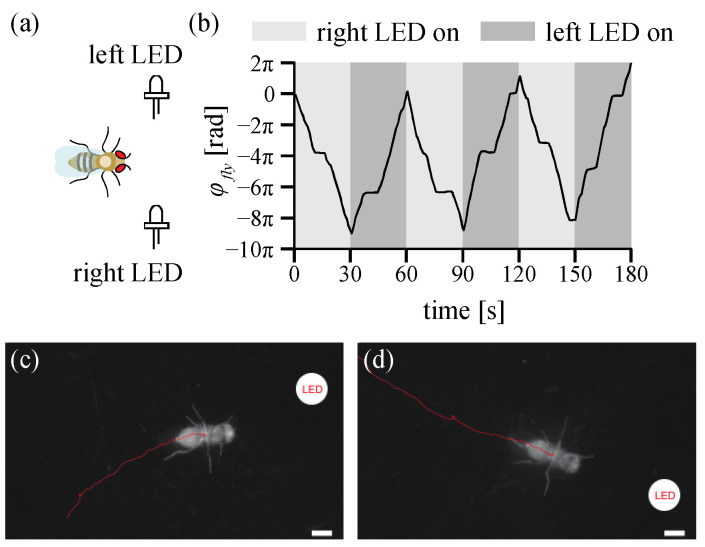
Phototaxis experiment setup and results (**a**) Schematics of the experimental setup. Broadband light LEDs are located on both the left and right sides. Lights on each side are toggled every 30 s. (**b**) The heading angle *ϕ_fly_* of the walking fly during the experiment. The fly showed a positive phototaxis. When the right LEDs are on, the fly moves toward the right (Clockwise rotation, the heading angle decreases). When the left LEDs are on, it moves toward the left (Counter-clockwise rotation, the heading angle increases). (**c**,**d**) The example NIR imaged during the experiment. The red color line is the path of the fly for the last one second. The white circle (added artificially) is the location of the LED. The scale bar is 1 mm. (**c**) NIR image at 49.4 s when the left LED is on. (**d**) NIR image at 74.7 s when the left LED is on. See Appendix A.

**Figure 10 sensors-21-01651-f010:**
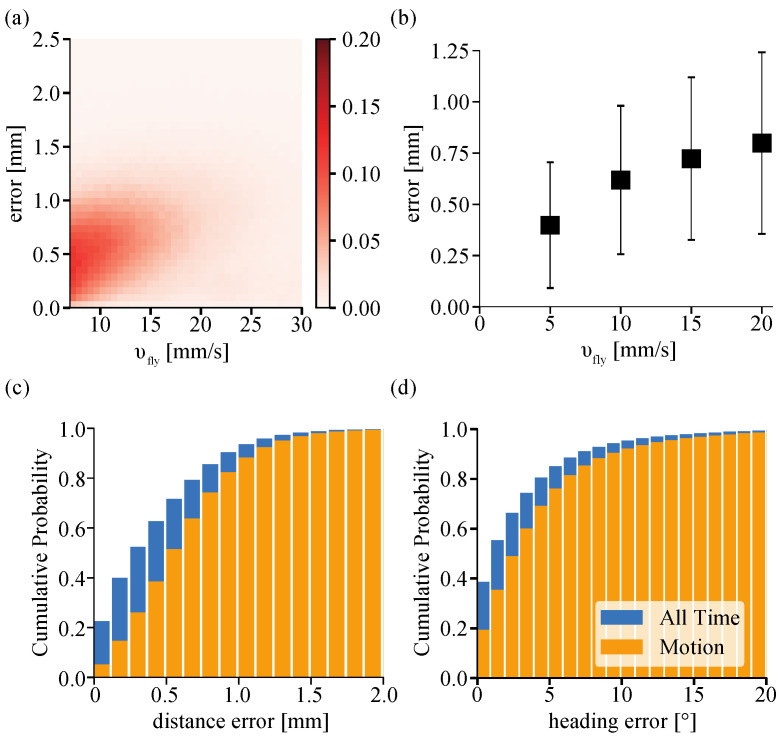
The performance of the TOLC. (**a**) The relationship between the distance error and the velocity of a walking fly. (**b**) The average error distance error for every 5.08 mm/s. The data point and bar represent the mean value and standard deviation. (**c**,**d**) The cumulative probability distribution of (**c**) the position error and (**d**) the heading error. The error data for entire experiments show as a blue bar, and those while a fly is in motion shows as an orange bar. We assumed that a fly is walking when its velocity is over >5.08 mm/s. The total experiment time is approximately 270 min across 20 flies and about 3.24 million images are processed in total.

## Data Availability

The data presented in this study can be provided by the corresponding authors upon reasonable request.

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
