# Peer review of "Navigation of a Freely Walking Fruit Fly in Infinite Space Using a Transparent Omnidirectional Locomotion Compensator (TOLC)"

_sensors, 2021, doi:10.3390/s21051651_

Round 1

Reviewer 1 Report

The authors are proposing a novel locomotion compensator setup adapted for drosophila. The main novelty from existing setup is the use of a transparent ball in order to track the drosophila movement from below the setup. This choice has the advantage to facilitate in situ brain recording by avoiding physical interference with a potential imaging or electrophysiology recording tool. In addition with presenting their setup, the authors provide behavioural data to demonstrate its efficiency.

The manuscript is well written and very easy to follow. The description of the setup and its functioning is clear. I have not noticed any major technical or conceptual issue.  My main concern is the very low sample size (sometimes even only one animal recorded on one trial) of the data provided to validate the setup. I do not think it would have been that time or work consuming to increase sample size to at least 20 animals in each condition to offer more representative data of the error rates, speed, …. I also wonder whether in behavioural tasks more complex than involving phototaxis, the transparent sphere might have a negative impact by reducing the ventral optic flow and visual feedback or by creating interference due to the potential perception by the insect of the wheels’ movements. More arguments against these possibilities should be provided if possible. Additionally, in the current version of the setup, a transparent ceiling is necessary to avoid the drosophila to fly off so how functional brain recording would be possible? Please specify.

Specific comments:

Figure 1 & 3: Providing a video would be useful to visualize the tracking systems and the smoothness of the ball movements due to the wheels.

  1. 151: I really appreciate this paragraph which offer convincing data on the negligible impact of distortion. However, such vague statement like ‘mostly staying’ should be avoided and replaced by objective numerical data (e.g % of time). This should be easy to provide in particular if you record the behaviour of more flies as I recommend.
  2. 217: please provide physical specification of the ‘moderate light intensity’, eg measured in W/m2 and the reference for the LED.
  3. 225: do you have explanation on why the angular velocity in the z-axis is smaller than on the other axis?

Paragraph 3.3: Recording and analysing data from only one fly is not acceptable. Please provide average data and dispersion from at least 20 walking paths of 5 minutes from different animals.

  1. 241: How do you define ‘normal activity’ to state that the speed recorded is ‘slightly less’ ? Please provide quantified data rather than vague appreciations.

Figure 8: While the error on the x and y axis seem indeed very low, at least on this single example…, the heading error of up to 15 ° is more problematic. I realize that, in this example, it is highly transient but data should be provided to allow to better evaluate the rate and duration of such error on a larger population of walking behaviours from different flies.

Paragraph 3.4: Idem, it is not possible to conclude anything convincing on the basis of a single experiment. Please test at least 20 flies.

Figure 9: please provide average distances between the fly eyes and the LEDs. Could you explain the ‘steps’ observables on the curve: are they due to pauses in the fly’s walking behaviour or due to the setup. Please also add data on the speed and distance travelled.

  1. 301: please increase sample size. Were these measurements in the absence of visual or olfactory stimulation? Please provide data in each condition.

Figure 10a: the figure is unclear or at least insufficient to represented the error rate. Please add a figure showing the relation between speed and error but without trying to combine with frequencies.

Reviewer 2 Report

I would like to congratulate the authors on the wonderful work. It is certainly a step in the right direction for conducting experiments on freely moving animals.
The presented method for tracking does add an additional option to the behavior experimenters. The idea of tracking animal using transparent spherical is definitely new and have not been reported before to the best of my knowledge. 

That being said I have the following comments for the authors, which may help them improve the quality of the paper and gain support from readers from a much wider community. I also have some questions which in my opinion should be answered in the text. Additionally, I find some information and citations missing which makes it difficult to evaluate the quality of work. Pls. refer attached document for detailed comments.

Round 2

Reviewer 1 Report

The authors have significantly improve their manuscript based on the comments of both reviewers. I recommend this manuscript for publication, even if I would have appreciated more behavioural data as requested in my first review. However, the authors made nevertheless the effort to include more data and I suppose they do not have the possibility to collect more data.